

# Characterization of Cold Land Hydrological Processes by Integrating In-Situ Snowpack Observations with a Land Surface Model in the Yellowstone River Basin, USA

Do Hyuk Kang[1]

[1]Weather Program Office, National Oceanic and Atmospheric Administration, Silver Spring, Maryland, 20910, U.S.A.

*Correspondence to*: Do Hyuk Kang (dk.kang@noaa.gov)

**Abstract.** In the eastern region of the North American Continental Divide in the upper Colorado Rockies, this study demonstrates that enhancing streamflow predictability from May to July in the Yellowstone River Basin is enabled. This streamflow improvement is achieved by employing a land surface hydrology model in the watershed, coupled with an updated

winter precipitation weather forcing dataset. Utilizing 13 snowpack telemetry stations from the US Department of Agriculture in the Yellowstone River Basin, the paper calculates ratios between a baseline simulated snowpack from the initial land surface model application and the observed snowpack. The average ratio serves as a constant multiplier for the existing snowfall weather forcing applied in the second land surface model simulation. As a result of the second simulation, the streamflow predictability reaches a Nash-Sutcliffe Efficiency (NSE) of 0.91, in contrast to the baseline simulation's 0.73 NSE during peak

streamflow periods. The study also explores cold land hydrological processes, particularly those related to snowmelt-driven streamflow. In addition to streamflow, two land surface variables such as snowpack and soil moisture are assessed against in-situ snowpack and satellite-based soil moisture observations in the Yellowstone River Basin. The comparisons reveal that the peak of soil moisture is mainly driven by springtime snowmelt and diminishes in the summer. The findings are confirmed by both land surface model simulations and satellite-borne soil moisture observations. Another noteworthy discovery is that soil

infiltration properties in the Yellowstone River Basin are wetter than the western Continental Divide in North America, resulting in amplified streamflow in the eastern side despite similar levels of snowmelt runoff on either side of the Continental Divide in the upper Colorado Rockies in the United States.

## 1 Introduction

Snowpack embraces significant importance in the upper Missouri River Basin situated in the western United States, as

highlighted in previous studies (Cherkauer and Lettenmaier 1999; Nasab and Chu 2023). Recent climate changes have brought increased public attention to this region, primarily due to factors like forest fires (Akinola and Adegoke 2019), severe drought (Woodhouse and Wise 2020), and floods triggered by atmospheric rivers (Lavers and Villarini 2013). Despite the extensive hydroclimatic research on the western side of the continental divide of North America (Leung et al. 1999; Quinn et al. 2004;



Nowak et al. 2012; Wanders et al. 2017), the eastern side has received comparatively less evaluations concerning hydrological
implications arising from recent climate change than the western side of the Continental Divide.

Climate anomalies, such as El Nino and La Nina, and their unexpected extremes have been identified as the main drivers of long-term drought in the upper Missouri River (Mauget and Upchurch 1999; Twine, Kucharik, and Poley 2005; Yeşilköy, Baydaroğlu, and Demir 2023). These hydrologic implications, including prolonged drought and extremely dry soil moisture,
often result in large-scale wildfires. Numerous studies have established a connection between drought and forest fires in the western part of North America (Littell et al. 2016; Collins et al. 2019; McElhinny 2020; Park, Cook, and Smerdon 2022). Additionally, severe forest fires have recently occurred in the eastern foothills of Northern Colorado (Stambaugh et al. 2006; Ward et al. 2006; Overpeck and Udall 2020; Hoell et al. 2020), resulting in unexpected air quality issues downstream of northern Midwest cities in the United States and Canada (Popovicheva et al. 2014; Kaulfus et al. 2017; Jaffe, Ninneman, and
Chan 2022).

This study focuses on one of the headwaters in the Upper Missouri River, namely the Yellowstone River Basin (YRB). The YRB is a snowmelt-dominant watershed, where the snowpack from the previous winter drives meltwater to the peak of streamflow in the summer of the following year. The paper assesses streamflow predictability influenced by snowmelt and
explores cold land hydrological processes, including snowpack and soil moisture, utilizing both in-situ and satellite-based observations.

Numerous studies have explored hydrological responses, particularly streamflow, in the upper Missouri River, with various researchers establishing this body of knowledge (Mauer and Lettenmaier 2004; Jha et al. 2004; Parajuli 2010; Qiao et al. 2014;
Vanderhoof, Christensen, and Alexandar 2019). In a recent study, Sando et al. (2022) employed a lumped hydrologic model to simulate year-round streamflow dynamics in the Missouri River Basin. They utilized the US Geological Survey hydrologic model and extended the existing modeling application from the Pacific Northwest to the Missouri River, demonstrating the model's robust applicability across diverse climatic and hydrologic regions.

However, the application of land surface hydrology models to the Yellowstone River, one of the headwaters of the Upper Missouri River, is relatively recent (Kannan et al. 2019; Flemming et al. 2021). While there is an increasing frequency of reports on flood events induced by snowmelt in the upper Missouri River basin (Olsen and Morton 2017; Woodhouse and Wise 2020), the understanding of hydrological processes in the headwater watershed remains limited. This study addresses this gap by applying a semi-distributed land surface hydrology model, specifically the Variable Infiltration Capacity (VIC)
model. The utilization of VIC in the Yellowstone River offers the additional advantage of assessing cold land hydrological processes, using simulated state variables such as snowpack, soil moisture, and streamflow driven by snowmelt. To validate





the accuracy of these simulated variables, comparisons are made against collocated in-situ observations (snowpack and streamflow) and satellite data (soil moisture) within the scope of this paper.

Climate change is also evident on the eastern side of the Colorado Rockies, pressing ongoing challenges in comprehensively understanding the entire cold hydrological process affected by both drought and altered streamflow due to climate variations. This paper presents a focused approach, conducting a prototype study in one of the headwaters of the upper Missouri River Basin—the Yellowstone River Basin. The study employs a land surface hydrology model, starting with available weather forcing data spanning from 1980 to 2021.


An identified issue in the available weather-forcing dataset is the consistent underestimation of snowfall forcing compared to the observed snowfall. To address this, the paper corrects the snowfall underestimation for an updated application of the land surface hydrology model. The study concludes by comparing the streamflow outputs between the baseline (without correction for winter precipitation) and the simulation adjusted for snowfall underestimation. In addition to streamflow assessments, the

study conducts an evaluation of the land model outputs using satellite-borne soil moisture observations, specifically NASA's Snow Active and Passive (SMAP), comparing them against the simulated soil moisture from the land surface hydrology model. Given that the primary source of soil moisture in the Yellowstone River Basin is derived from the snowpack and its melt, this comparison against satellite observations demonstrates insights into the cold land hydrological processes occurring in the headwaters of the upper Missouri River.


The structure of this paper is outlined as follows: In Section 2, we provide an overview of the Yellowstone River Basin (YRB). We introduce the land surface hydrology model, specifically the Variable Infiltration Capacity (VIC) model, along with details about the weather forcing dataset. Additionally, we explain the method used to adjust winter precipitation, using the ratios between data from 13 SNOTEL snowpack observations and the baseline application of the land surface model within the YRB.

The locations of streamflow observations are detailed, situated in Livingstone and Billings, Montana, USA. Two other key state variables, soil moisture, and snowpack, are also discussed. The validation process is subject to observations for soil moisture using NASA's Soil Moisture Active Passive (SMAP) and snowpack using the Advanced Microwave Scanning Radiometer (AMSR). Moving to Section 3, we present the results, focusing on the enhanced streamflow achieved with adjusted winter precipitation forcing and the simulated soil moisture in the YRB. Finally, in Section 4, we conclude by summarizing

the key findings of this paper, demonstrating the cold land hydrological processes on the eastern side of the upper Colorado Rockies within the Yellowstone River Basin



## 2 Methods and Materials

### 1.1 Study Area

The Yellowstone River, one of the primary headwaters of the upper Missouri River, originates from the Absaroka Range and converges with the Missouri River at the Montana-North Dakota border in the United States. Notably, it holds the distinction of being the longest free-flowing river in the lower 48 U.S. states (Eddy-Miller and Chase 2015). The river's source begins on the slopes of Yellowstone Lake, standing at an elevation of 3,700 meters above sea level (asl). To facilitate the application of the land surface hydrology model, a 0.125-degree grid cell is prepared using digital elevation models. This results in the

identification of 289 digital elevation grid cells that cover and delineate the watershed, with the outlet situated in Billings, Montana, US.

The Trout Peak, also at an elevation of 3,700 meters asl, marks the southern edge of the watershed and serves as the source of the Yellowstone River. To the northeast of the watershed, the Custer Gallatin National Forest extends, featuring its highest

peak, Granite Peak, reaching 3,900 meters asl. The flow of the river is directed initially northward and then eastward toward the outlet location. Two streamflow gauges are selected to this study: 1) an upstream gauge and 2) another located at the outlet of the Yellowstone River Basin. The upstream gauge, named 'Yellowstone River near Livingston, MT – USGS 06192500,' is maintained by the US Geological Survey (USGS) and positioned at an altitude of 1,587 meters asl. Meanwhile, the streamflow gauge at the outlet, named 'Yellowstone River at Billings MT- USGS 06214500,' is also maintained by the USGS and is located

at the outlet point of the entire headwaters, with an altitude of 992 meters asl. Detailed descriptions of these streamflow gauges are summarized in **Table 1**.

**Table 1. Summary of USGS streamflow gauge in Billings MT of the Yellowstone River Basin.**

| Metadata Element | Location Metadata | Location Metadata |
|---|---|---|
| Agency | U.S. Geological Survey | U.S. Geological Survey |
| Site ID | 06192500 | 06214500 |
| Site name | Yellowstone River Near Livingston, MT | Yellowstone River At Billings, MT |
| Site type | stream | stream |
| Decimal latitude | 45.59 | 45.80 |
| Decimal longitude | -110.56 | -108.46 |



| Range of daily values | 05/01/1897~Present | 10/01/1989~Present |
|---|---|---|

## 2.2 In-situ snowpack observations

In addition to the streamflow gauges, a network of snowpack monitoring stations is essential for methodology and evaluation of the study. The US Department of Agriculture maintains an observation network known as 'SNOTEL' (Snow Telemetry) systems in the western United States (Schaefer and Paetzold 2001). Within the basin, a total of 13 SNOTEL stations have been identified, and their locations are depicted in **Figure 1** and their details are described in **Table 2**. Eleven of these stations are situated in the mountains surrounding Granite Peak, while the remaining two are positioned in the north-western foothills of Crazy Peak within the Custer Gallatin National Forest.

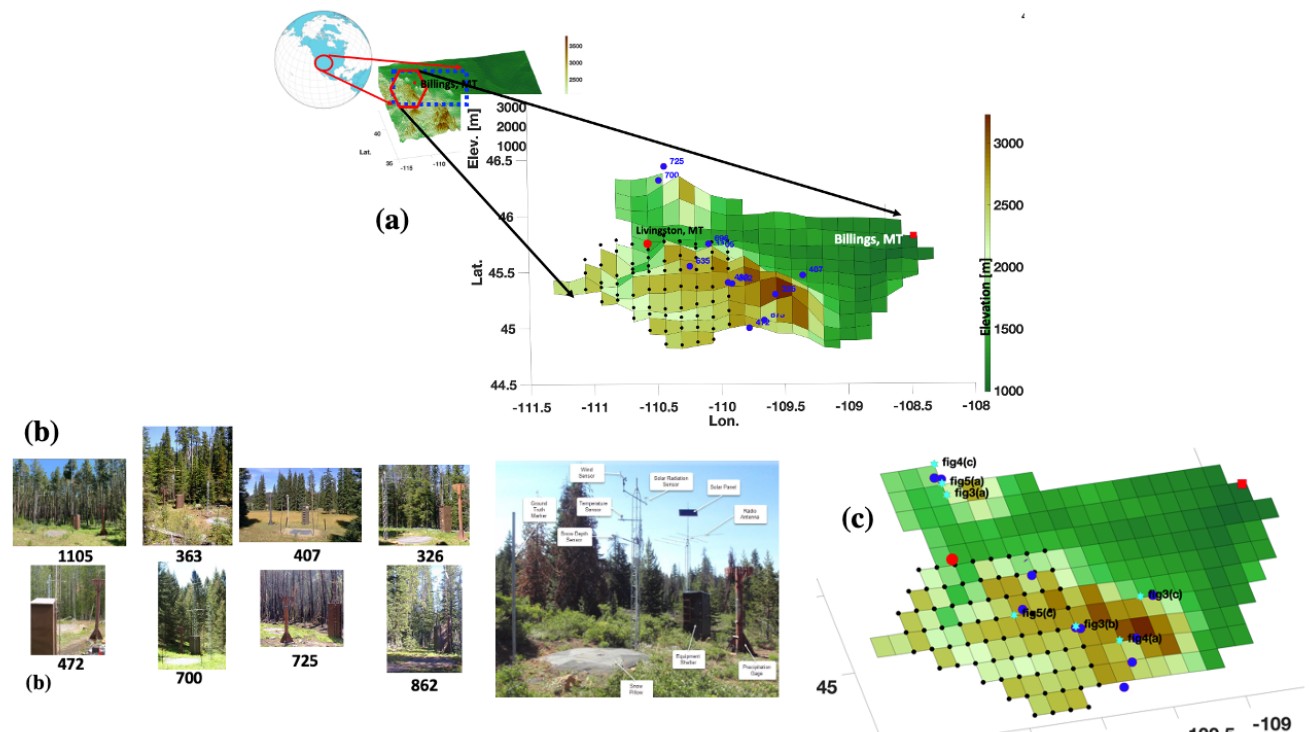

**Figure 1:(a) Location of the upper Missouri River Basin of the United States, the Yellowstone River Basin (a blue dotted square), the western tributary of the upper Missouri River, and its headwaters (a red hexagon) represented by 0.125-degree digital elevation models with 233 grid cells where the outlet is located in Billings, Montana, US. (b) Snapshots of the 13 SNOTEL sites in the basin and a schematic view of SNOTEL sensor deployments. Pictures from the US Department of Agriculture SNOTEL website. (c) locations of the Figure 3-8 in the basin.**





**Table 2. List of USDA SNOTEL sites used in the Yellowstone River Basin.**

| SNOTEL ID. | Lat. | Lon. | Elevation [m] |
| --- | --- | --- | --- |
| 1105 | 45.50 | -110.08 | 1930 |
| 696 | 45.42 | -110.09 | 2691 |
| 363 | 45.27 | -110.25 | 2033 |
| 635 | 45.2 | -110.24 | 2697 |
| 407 | 45.19 | -109.35 | 2392 |
| 480 | 45.06 | -109.94 | 2773 |
| 862 | 45.05 | -109.91 | 2651 |
| 670 | 45.01 | -110.01 | 2240 |
| 326 | 44.94 | -109.57 | 2852 |
| 875 | 44.80 | -109.66 | 2331 |
| 472 | 44.65 | -109.78 | 2804 |
| 725 | 46.09 | -110.43 | 2466 |
| 700 | 46.11 | -110.47 | 1981 |

The snowpack, accumulated during the winter of the preceding year, contributes to the annual peak of streamflow in the
Yellowstone River in the following year after the melt. Year-round streamflow observations are conducted at the upstream
gauge in Livingston, MT, and at the outlet gauge in Billings, MT. Since the snowpack (specifically, the snow water equivalent,
SWE) is simulated by the land surface hydrology model, the study compares the collocated simulated SWEs against the
observed SWEs measured at each corresponding SNOTEL station. This comparison serves as a quantitative measure of the
underestimated amount of snowfall simulated by the land surface hydrology model, driven by the weather forcing dataset, in
this case, the ERA 5 (ECMWF atmospheric reanalysis).

**2.3 Land Surface Model**

Since the pioneering development of the Variable Infiltration Capacity (VIC) model by Liang et al. in 1999, it has demonstrated
widespread application in various watersheds globally. Notable regions include North America (Mote et al. 2005; Luo and
Wood 2007; Bohn and Vivoni 2019), Asia (Xie et al. 2007; Zhang et al. 2013), and Europe (Bohn et al. 2007; van Vliet et al.
2015; Greuell et al. 2018). VIC, particularly in the context of snowmelt runoff processes, has been previously applied in the
Missouri River Basin (Cherkauer and Lettenmaier 1999; Tavakoly et al. 2017; Badger et al. 2018). In response to the
unprecedented climate changes observed recently, this paper represents a timely effort to assess the cold land hydrological



processes in the headwaters of the upper Missouri River—the Yellowstone River. Furthermore, the Yellowstone River is important as an unregulated river without any water structures, allowing the VIC model application to independently evaluate

streamflow contributions from the preceding snowpack in the previous year across the watershed.

The Yellowstone River Basin has a watershed area of approximately 55,000 km2, divided into 289 spatial grid cells with a resolution of 0.125° in the VIC grid. Two main headwaters originate from the southwest of the mountain ranges with the Granite and Pilot Peaks, and from the northwest mountains with the Iddings and Crazy Peaks. These two primary stems

converge at the outlet of the Yellowstone River Basin in Billings, Montana, USA. The VIC model is configured with 18 rows and 29 columns, representing elevation, flow direction, and the outlet grid cell drives the routing model after running the VIC with the weather forcing dataset.

Similar to the preceding study by Kang and Jung (2023), the land surface model is categorized into two groups: the baseline

VIC simulation ($VIC_{BL}$) and the multiplication factor ($M_f$)-adjusted simulation ($VIC_{MF}$). The multiplication factor ($M_f$) is determined by averaging a scatter plot of the ratios between measured snow water equivalents (SWEs) from SNOTEL and the SWEs from VIC simulations. This multiplication factor ($M_f$) spans a range of about 0.5 to 3.5 based on $VIC_{BL}$, determined using the minimum and maximum elevations in the watershed. The average $M_f$ is calculated to be 1.6. Another condition is introduced where (Tmin + Tmax)/2 < -4 °C instead of 0.0° C, requiring that $M_f$ is applied only when the daily air temperature

is cold enough at the grid cell.

Aside from the adjustment to the precipitation forcing through $M_f$ and the change in the weather forcing dataset, the setup involving VIC and ROUT is equivalently applied to both simulation sets ($VIC_{BL}$ and $VIC_{MF}$). This ensures a consistent assessment of cold land hydrological processes in the Yellowstone River Basin across the two simulation scenarios.


For streamflow calibration, a classical method outlined by Lohmann et al. (1998) is employed for the years 2000 to 2010. This method involves adjusting five soil properties to optimize the fit between the simulated streamflow from VIC and the observed streamflow data from USGS during this period. Subsequently, the following 10 years (2011 to 2020) are used to validate the streamflow performance based on the determined five soil parameters. In the case of two sets of VIC applications, namely

$VIC_{BL}$ and $VIC_{MF}$, the same streamflow calibration procedure is applied to achieve optimal streamflow performance by minimizing the difference between USGS observed streamflow and VIC simulated streamflow. This independent comparison between $VIC_{BL}$ and $VIC_{MF}$ serves to evaluate the effect of the multiplication factor ($M_f$) using a comparison between simulated and observed Snow Water Equivalent (SWE) in the Yellowstone River Basin.



## 2.4 Weather Forcing Dataset

The ERA-5 reanalysis dataset, a product of the European Centre for Medium-Range Weather Forecasts (ECMWF) (Hersbach et al., 2020), was employed to drive the Variable Infiltration Capacity (VIC) model. The ECMWF has been generating atmospheric reanalysis datasets available from 1980, and for this study, the recently launched fifth generation, ERA-5, was used. This dataset provides hourly air temperature, precipitation, and wind speed information, and after downloading, these variables are processed to derive four main atmospheric variables for the VIC model. These include daily maximum air

temperature (Tmax, °C), daily minimum air temperature (Tmin, °C), daily precipitation (PREC, mm), and wind speed (WS, m/sec) spanning from 1980 to 2020. Notably, the daily precipitation variable (PREC) is adjusted with a multiplication factor ($M_f$), determined from a comparison between observed Snow Water Equivalents (SWEs) and the corresponding baseline VIC simulated SWE ($VIC_{BL}$).

**2.5 Remotely retrieved soil moisture and snow water equivalent observations**

This study validates hydrologic simulations based on two key observations: soil moisture, SWE, and streamflow. The VIC model simulates soil moisture and SWE, while the ROUT application simulates streamflow. Streamflow accuracy is assessed at two USGS streamflow gauge points in Livingstone and Billings, MT, USA. However, due to spatial and temporal variations in soil moisture across the Yellowstone River Basin, spatio-temporal observations are necessary. Satellite-derived soil moisture

data from the NASA Soil Moisture Active and Passive (SMAP) satellite are utilized for evaluation, covering the period since 2015. $VIC_{BL}$ and $VIC_{MF}$ independently simulate soil moisture, and their outputs are compared against SMAP observations. The VIC grid has a resolution of 0.125 degrees, approximately 14 km, while the 36 km resolution of SMAP observations can encompass four VIC grid cells. The default comparison involves depth-averaged simulated soil moisture against corresponding SMAP values. In cases of deep (above 200 mm SWE) and shallow (below 200 mm SWE) snow water equivalent (SWE),

three-layered soil moisture simulations are compared against a single SMAP value.

Furthermore, for an independent comparison of snow water equivalent (SWE) which is also varying in spatial-temporal domains, additional SWE observations are obtained from the Advanced Microwave Scanning Radiometer (AMSR) as obtained from Tedesco and Jeyaratnam (2019). A 25 km grid of SWE in millimeters is extracted from the AMSR dataset. Subject to

passive microwave radiometry theory, AMSR provides SWE observations that are utilized to validate the simulated SWE within the basin.



## 3 Results and discussions

Following the methodology of the prior study conducted by Kang and Jung (2023), the results section of this research assesses
hydrologic outputs, specifically focusing on three state variables: 1) snowpack, 2) soil moisture, and 3) streamflow. Initially,
the simulation of snow water equivalent with baseline weather forcing is performed using $VIC_{BL}$, and the outcomes are then
updated with those derived from the $VIC_{MF}$ model, utilizing the $M_f$ adjusted weather forcing dataset. The accuracy of the
simulated SWEs from $VIC_{BL}$ and $VIC_{MF}$ is validated against SNOTEL SWE observations. Subsequently, satellite-derived soil
moisture is employed to evaluate the simulated soil moisture within the watershed generated by $VIC_{BL}$ and $VIC_{MF}$. Lastly, the
enhancement in streamflow achieved by $VIC_{BL}$ and $VIC_{MF}$ is evaluated through a comparison of the simulation results with
two USGS streamflow observations. This study provides a comprehensive evaluation of cold land hydrological processes in
the Yellowstone River Basin, employing dynamic simulations of snowpack, soil moisture, and streamflow within the
watershed.

### 3.1 Performances of snow water equivalent estimation

**Figure 2** presents comparisons of $VIC_{BL}$, $VIC_{MF}$ SWEs against SNOTEL SWE observations. The $M_f$ value of 1.60 is
determined by the ratio between SNOTEL SWE and $VIC_{BL}$ simulated SWE, utilizing 13 ratio values to calculate the final $M_f$.
This $M_f$ is then applied to regenerate the weather forcing dataset, specifically for winter precipitation, resulting in a new set of
SWE from $VIC_{MF}$. The observed snowfall underestimation, a well-documented phenomenon in the literature, is affirmed in
this study due to the utilization of the European Reanalysis Assimilation 5 (ERA-5) weather forcing dataset (Hersbach et al.,
2016). Many weather reanalysis datasets heavily rely on surface-based weather observations, leading to the issue of "snow
undercatch" as discussed in previous studies (Fassnacht, 2004; McDonald and Pomeroy, 2007). This paper builds upon the
established understanding of snow undercatch, using it to update the winter precipitation forcing dataset and subsequently
simulate snow water equivalent, soil moisture, and streamflow, particularly in the case of $VIC_{MF}$. Section 2.3 provides detailed
information on the methodology for updating winter precipitation based on $M_f$ in the basin.

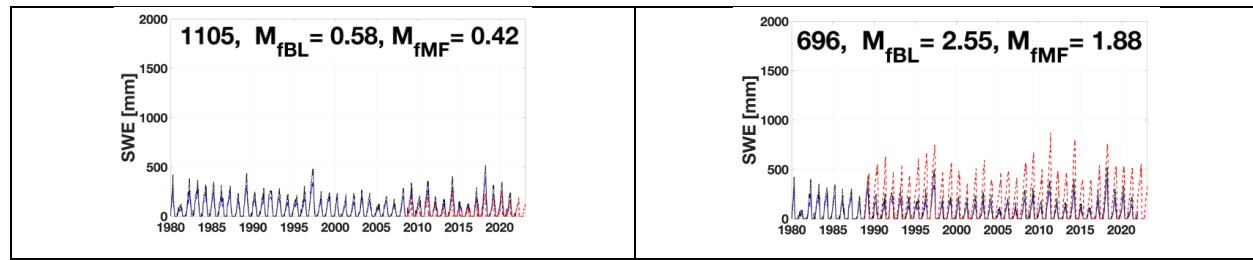








**Figure 2: The comparison of snow water equivalents between SNOTEL observations and the two VIC simulations (VIC$_{BL}$ and VIC$_{MF}$).**

## 3.2 Soil moisture simulation induced by snowfall update

Produced by VIC$_{MF}$, this section outlines the updated soil moisture simulations. The simulated soil moisture outputs are

assessed against observed soil moisture data obtained from satellite-based measurements by the NASA SMAP satellite spanning the period from 2015 to 2021. **Figure 3** provides a visual representation of soil moisture at three selected observation points denoted as the 'west (near SNOTEL 725, 2468 m asl),' 'middle (near SNOTEL 480, 2773 m asl),' and 'east (near SNOTEL 407, 2392 m asl)' regions of the watershed. These points are strategically chosen to capture localized variations in soil moisture influenced by snowmelt, and their positions are indicated in **Figure 1a&c**. To facilitate comparison with NASA

SMAP observations, the three-layered soil moisture simulations generated by VIC are averaged to yield a single value. Zoomed plots for the 2016 water year are presented in **Figures 3b, 3d, and 3f**, covering the period from October 2015 to September 2016. Additionally, the corresponding SWE simulations (VIC$_{MF}$) are plotted with a reversed direction on the y-axis.

The enhancements in soil moisture simulations can be attributed to the improved predictability of snow water equivalent

(SWE). As illustrated in (**Fig 3a, c, e**), VIC$_{MF}$ SWEs exhibit a strong fit with SWE observations. The application of the multiplication factor (M$_f$) to the VIC$_{MF}$ weather forcing dataset ensures that VIC$_{BL}$ SWEs tend to underestimate the actual SWE. This underestimation, in turn, results in underestimated soil moisture levels (indicated by the green-dotted line in VIC$_{BL}$ soil moisture) or causes a temporal delay in the release of soil moisture due to reduced SWE associated with lower winter precipitation.


As the snow water equivalent (SWE) increases and reaches its peak, particularly until May and even June (as depicted in **Fig 3b, d, f**), the corresponding soil moisture experiences a peak that aligns with the SWE peak with a temporal delay. Subsequently, the soil moisture diminishes as the SWE gradually disappears. The soil moisture simulations at points in the west and middle regions of the watershed (**Fig 3b, d**) closely match the SMAP observations. However, in the eastern part of

the watershed (**Fig 3f**), VIC$_{MF}$ overestimates the soil moisture simulation. Examining **Figure 1c**, it is evident that the point in the east is located in the foothill of the mountain, where streamflow is more likely to flow rapidly. This discrepancy in the simulation is attributed to the fact that the swift attenuation of soil moisture is likely and reflected in the SMAP observations but not adequately captured in the VIC$_{MF}$ simulation. Overall, across the west, middle, and east regions of the basin, both SWE and soil moisture simulations align well with in-situ SWE and satellite-based soil moisture observations.


**Figures 4(ab)** and **4(cd)** provide a comparative analysis of snowmelt-soil moisture processes at high (near SNOTEL 326, 2800 m above sea level) and low (near SNOTEL 700, 1900 m above sea level) elevations. In **Figure 4(ab)**, where SWEs for both





SNOTEL and VIC$_{MF}$ surpass 400 mm, the soil moisture peaks up to 0.3 cm³/cm³ for both VIC$_{MF}$ simulation and SMAP
observation. The third-layer simulation from VIC$_{MF}$ more accurately captures the peak of the SMAP observation, whereas the
depth-average soil moisture simulation falls short. In **Figure 4(cd)**, where both simulated and observed SWEs only reach 200
mm, the soil moisture peak associated with snowmelt aligns well with the average soil moisture simulation against the SMAP
observation. However, during the dry season from August to October 2021, the third-layer soil moisture is effectively
represented in comparison with the SMAP observation. This suggests that the SMAP L-band radiometer/radar signal can
deeply penetrate the soil when it is not saturated with liquid water (Schwank et al., 2004). Consequently, the third-layer soil
moisture simulation precisely reflects realistic conditions at lowland elevations where SWE does not exceed 300 mm during
the dry season.

Another set of satellite-based SWE products is included with a passive microwave observation for SWE, the Advanced
Microwave Scanning Radiometer-Earth (AMSR-E or AMSR-2) (Kelly et al. 2003), shown in **Figure 5**. Microwave saturation
effect above 200 mm SWE is confirmed by comparing two sites with SWE under (**Fig. 5ab**) and above (**Fig. 5cd**) 200 mm,
where the SWE estimation from AMSR is challenging with the SWE above 200 mm (Vuyouvich et al. 2014). However,
SNOTEL SWE observations satisfy the amount of the VIC$_{MF}$ simulated SWE above 200 mm. Therefore, VIC$_{MF}$ soil moisture
can capture a trend of the observed SMAP soil moisture, while VIC$_{BL}$ soil moisture underestimates the SMAP observation,
shown in the water year 2021 in **Figure 5cd**. The advantage of low AMSR SWE (Kang, Lee, and Kim 2022) was addressed
in the Red River Basin of the North where the SWE is shallow mostly under 200 mm. In the Red River Basin, AMSR-E SWE
is used for the update of the snowfall weather-forcing to improve streamflow predictability. In this alpine watershed where
SWE often exceeds 200 mm, the AMSR SWE is not appropriate to correct the underestimation of the winter snowfall. Instead,
the in-situ SWE observations such as the SNOTEL apply to the deep snowpack above 200 mm SWE to update the snow
weather forcing, thus enhancing the streamflow predictability.

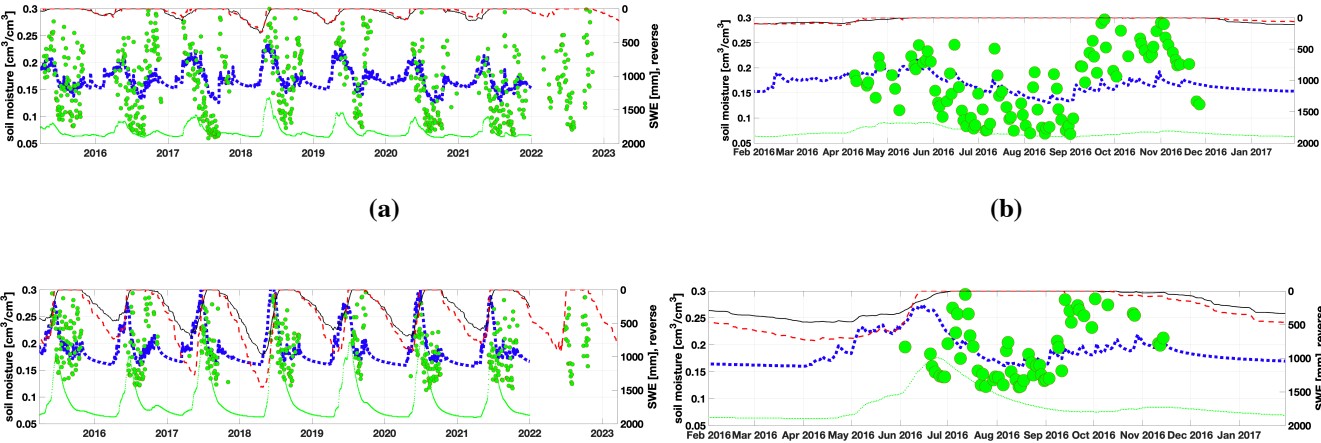

(a)                                (b)



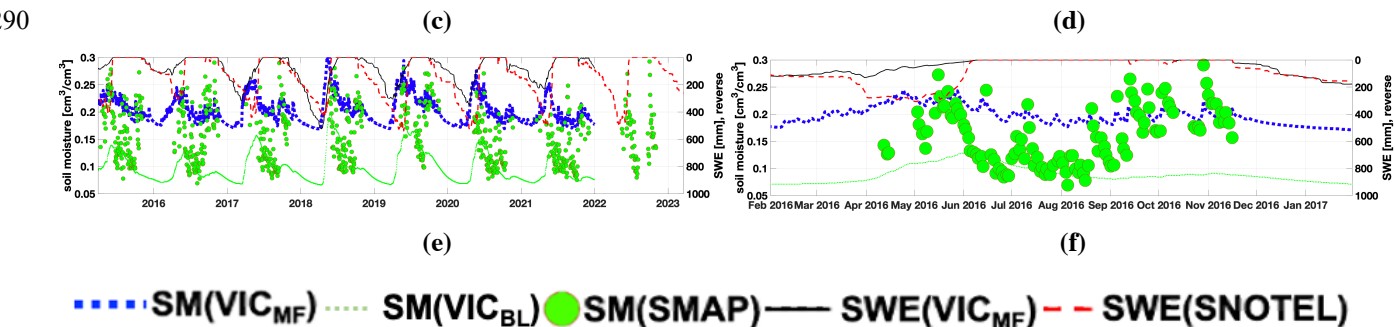

**Figure 3: The comparison of soil moistures and snow water equivalents between modeling and observations at three selected points, representing the (a,b) west (SNOTEL 725), (c,d) middle (SNOTEL 480), and (e,f) east (SNOTEL 407) regions of the watershed for years 2015-2021. (a,c,e) Black boxes indicate (b,d,f) their zoomed view of the 2017 water year.**

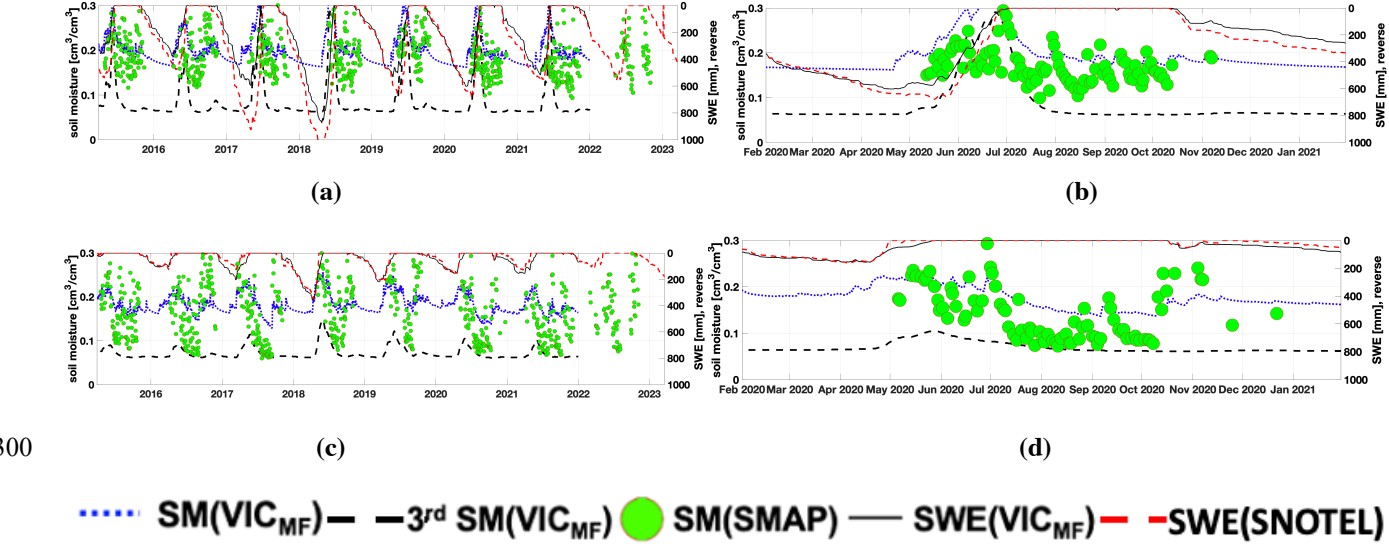

**Figure 4: The comparison of soil moistures and snow water equivalents between modeling and observations at two selected points (a,b) near the top of the mountain and (c,d) in the relatively low land, adjacent to the SNOTEL (326 – 2800 m a.s.l., 700 – 1900 m a. s. l.). (a,c), and (b,d) is their zoomed view of the 2021 water year.**

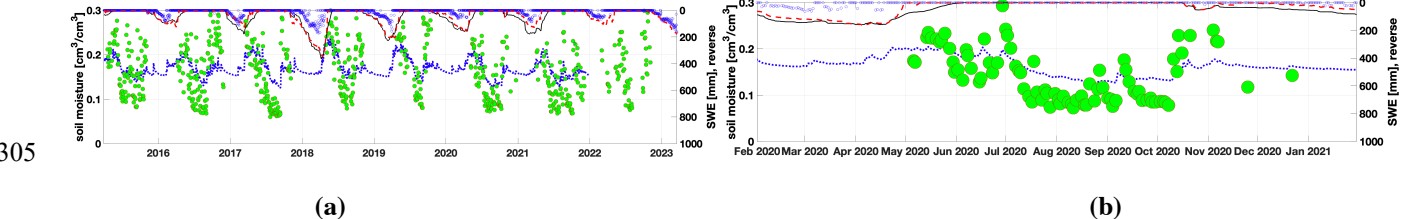





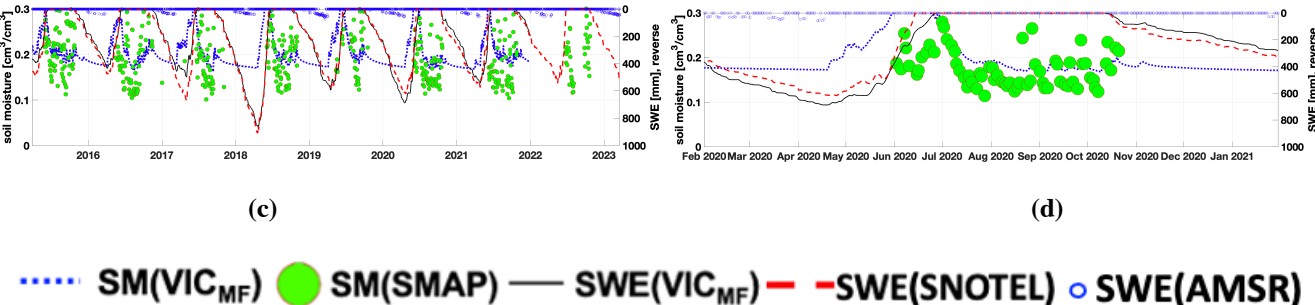

**(c)** **(d)**

····· SM(VIC_MF)  ● SM(SMAP)  —— SWE(VIC_MF)  – –SWE(SNOTEL)  ○ SWE(AMSR)

**Figure 5: The comparison of soil moistures and snow water equivalents between modelling and observations at two selected points (a,b) in the northern mountain with SWE values below 200 mm adjacent to the SNOTEL 725 (2400 m a.s.l.), and (c,d) in the southern mountain, adjacent to the SNOTEL 635 (2600 m a.s.l.), with SWE values above 200 mm. (a,c). Their zoomed views of the 2021 water year are in (b,d). AMSR SWE is also included.**

### 3.2 Influence of snow water equivalent on streamflow

For the validation against two streamflow gauges, two sets of VIC simulations (VIC_MF and VIC_BL) are generated. Both VIC_MF and VIC_BL are treated with the same calibration procedure from 2000 to 2010, with independent validations conducted from 2011 to 2020. The observed streamflow measurements at the upstream (Livingston, MT) and outlet (Billings, MT) are used for calibration/validation. **Tables 3 and 4**, along with **Figure 6**, present the Nash Sutcliffe Efficiency (NSE) values as performance measures.

In **Table 3,** for the upstream, VIC_MF (0.86) during calibration is outperformed by VIC_BL (0.58), but during the validation period, it exhibits a better NSE (0.73) than VIC_BL (0.54). Focusing on the snowmelt-season (April to July), VIC_MF achieves an NSE of 0.79, surpassing VIC_BL's 0.50. This indicates that VICMF performs better in streamflow driven by snowmelt in the spring season in the upstream. **Table 4** also shows that VIC_MF outperforms VIC_BL in both calibration (0.91 vs. 0.82) and validation (0.65 vs. 0.63), as well as during peak months (0.91 vs. 0.73). In the period from 2004 to 2010, VIC_MF achieves an NSE of 0.92, while VIC_BL only reaches 0.80. Overall, the impact of snowfall updates (VIC_MF) is more pronounced at the outlet than in the upstream. This is attributed to a more substantial update of snowfall occurring across the entire watershed, contributing more significantly to the outlet point than to the upstream.

**Figure 7** illustrates the monthly average streamflow at the upstream (bottom) and the outlet (top). The overall streamflow amount is larger at the outlet compared to the upstream. A streamflow peak in June from VIC_MF at the outlet aligns more closely with the observed streamflow than the peak at the upstream. The impact of the snowfall updates is more pronounced at the outlet point than at the upstream, showcasing a more consolidated effect. The improved NSE values at the outlet, as reflected in Table 4, further highlight the positive effects of the updated snowfall in enhancing streamflow simulations.





**Figure 8** provides a summary of small, medium, and large amounts of streamflow in different water years—specifically, in 2002, 2008, and 2018. The left panels encompass the entire watershed, while the right panels focus on the upstream. With increasing streamflow, a closer match between $VIC_{MF}$ and USGS observations is more probable. This alignment is attributed to the substantial impact of snowmelt-driven streamflow. Consequently, the update of winter precipitation enhances the

predictability of streamflow, particularly during the peak season in the Yellowstone River Basin.

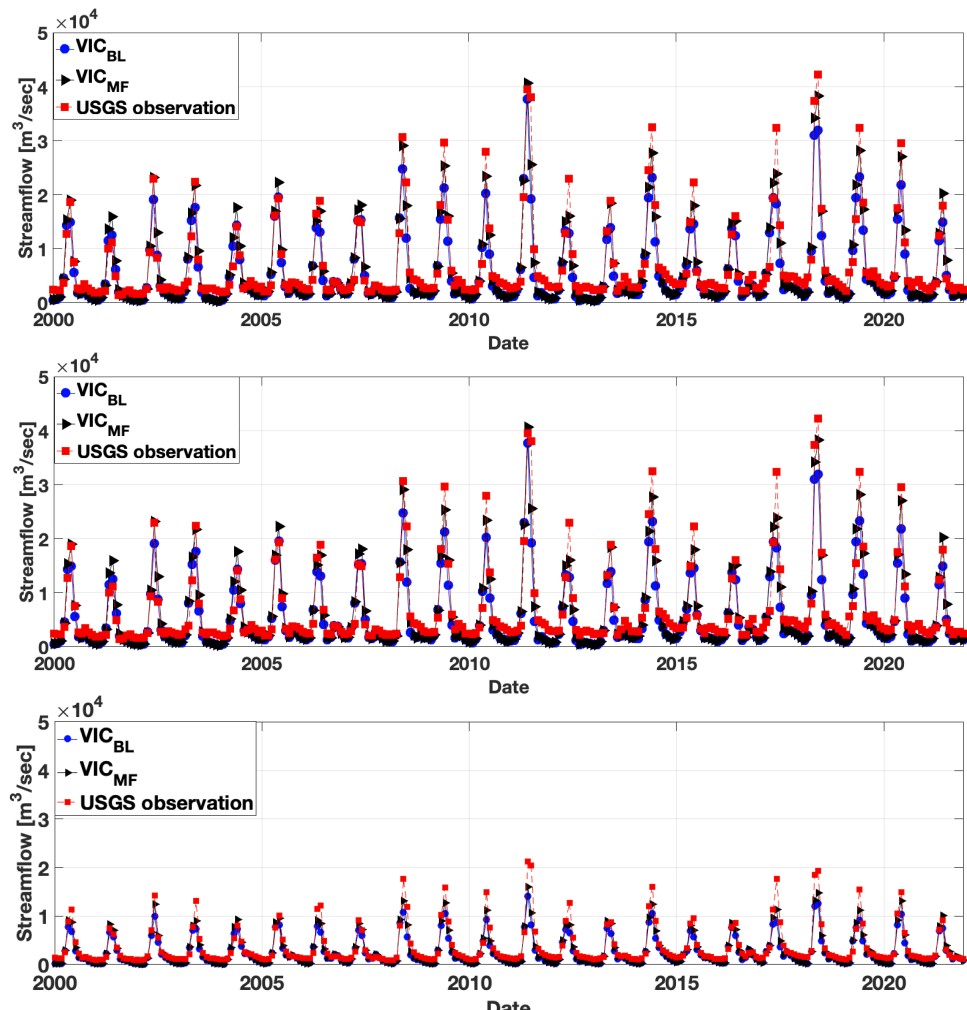

**Figure 6: Monthly time series of streamflow rates at the upstream (top) and at the outlet (bottom) of the Yellowstone River Basin.**



**Table 3. Evaulation index of Nash-Sutcliffe Efficiency coefficients for baseline (VIC$_{BL}$) and Mf-adjusted VIC simulations (VIC$_{MF}$) near Livingston, MT.**

| Period | Calibration (2000-2010) | Validation (2011-2021) | Selective in Cal. 2004-2010 | Peak streamflow months (April-July) 2006-2010 |
|---|---|---|---|---|
| VIC$_{BL}$ | 0.5815 | 0.5417 | 0.5415 | 0.5078 |
| VIC$_{MF}$ | 0.8615 | 0.7340 | 0.8463 | 0.7937 |

**Table 4. Evaluation index of Nash-Sutcliffe Efficiency coefficients for baseline (VICBL) and M$_f$-adjusted VIC simulations (VIC$_{MF}$) in Billings, MT.**

| Period | Calibration (2000-2010) | Validation (2011-2021) | Selective in Cal. 2004-2010 | Peak streamflow months (April-July) 2006-2010 |
|---|---|---|---|---|
| VIC$_{BL}$ | 0.8278 | 0.6301 | 0.8007 | 0.7367 |
| VIC$_{MF}$ | 0.9194 | 0.6596 | 0.9244 | 0.9115 |

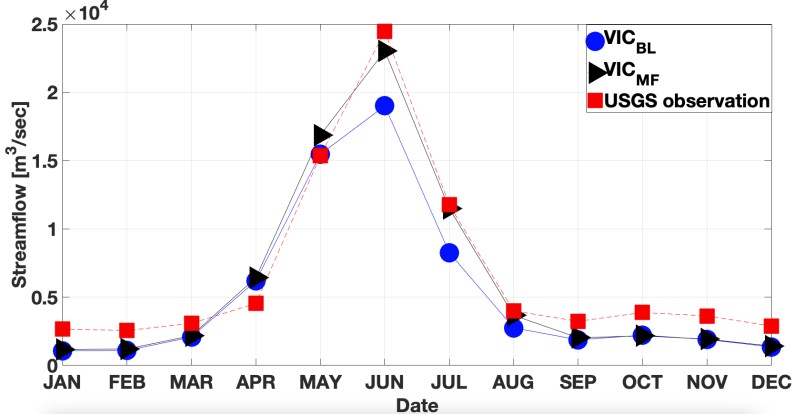

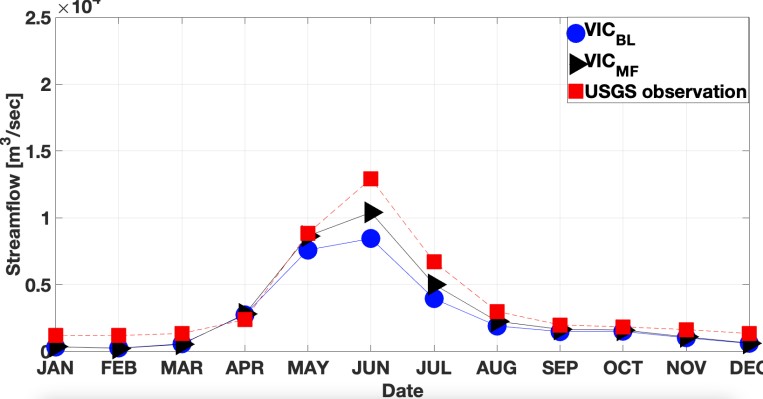





**Figure 7: Average monthly streamflow rates at outlet (top) and upstream (bottom) points from the baseline (VIC$_{BL}$) and M$_f$ adjusted VIC simulations (VIC$_{MF}$) along with USGS observations for water years 2000-2021.**


**Figure 8: Average monthly streamflow rates from the two VIS simulations and USGS observations for a specific water year (a) 2001-2002, (b) 2007-2008, and (c) 2018-2019 where (left) is for the entire basin and (right) is for the upstream.**

## 3.3 Comparison of snowmelt driven streamflow between east and west sides of North American Continental Divide

Kang and Jung (2023) conducted a comparable study on cold land hydrological processes in the headwaters of the Snake River Basin, located on the west side of the Continental Divide, in contrast to the eastern side of the Continental Divide for the Yellowstone River Basin. Notably, the predictability of peak streamflow in May and June is more advanced in the Yellowstone



River Basin (YRB) compared to the headwaters of the Snake River Basin (SRB). Despite similar watershed sizes, with 327 grid cells for SRB and 237 grid cells for YRB, the peak streamflow in YRB exceeds 250,000 m³/sec, whereas SRB only
reaches a peak of 350 m³/sec. This disparity may be associated with differences in precipitation patterns between the western and eastern sides of the Continental Divide in the upper Colorado Rockies. It is possible that rainfall during the snowmelt season in YRB exceeded that in SRB. However, annual precipitation is comparable, averaging 35 mm and 44 mm per month in Idaho Falls, ID, and Billings, MT, respectively, according to the ECMWF database. Hence, another factor must influence the streamflow difference between the two watersheds.


**Figure 9** illustrates a contrast in soil characteristics between 'xeric aridic' for the Snake River Basin and 'typic ustic' for the Yellowstone River Basin. 'Xeric aridic' represents a typical dry soil, while 'typic ustic' falls in the middle, exhibiting characteristics between dry and wet soils (Baillie 2001 from the US Department of Agriculture). This suggests that infiltration from snowmelt water is impeded in the Yellowstone River Basin compared to the Snake River Basin due to the wetness of the
soil surface. As a result, it is recommended to account for different infiltration characteristics when applying land surface hydrology models between the Continental Divide of the upper Colorado Mountains.

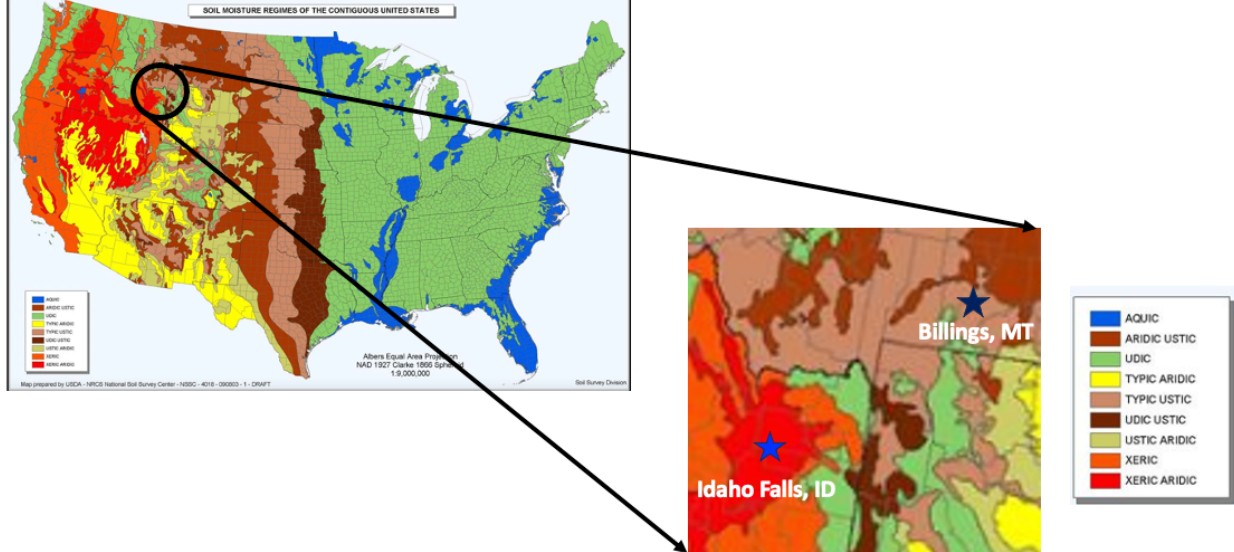

**Figure 9: Soil classification map in the United States and its zoom of the Snake River Basin and the Yellowstone River Basin. (United States Department of Agriculture Soil Survey Staff accessed in 2023; Baillie 1999).**




## 3 Concluding remarks

The study investigates cold land hydrological processes in the Yellowstone River Basin (YRB) through the application of the semi-distributed land surface hydrology model, the Variable Infiltration Capacity (VIC). The research employs a weather forcing dataset obtained from ECMWF ERA-5, and snowpit observations collected in situ are utilized to correct the underestimation of winter precipitation. Similar to the work conducted by Kang and Jung (2023), this paper concentrates on the assessment of snow water equivalent, soil moisture, and streamflow on the eastern side of the Continental Divide in the upper Colorado Mountains in North America. Once again, the study demonstrates an improvement in the predictability of streamflow, particularly during peak periods, through the application of the updated winter precipitation forcing in the VIC model for the YRB. The key conclusions are drawn point by point below.

- The update of the winter precipitation using in-situ snowpack observations increases a predictability of the peak streamflow in May and June of the streamflow in the upstream and outlet of the Yellowstone River Basin

- Changes of soil moisture in the YRB are well captured by satellite-borne SMAP soil moisture observations from a peak to decline associated with the snowmelt from May to October.

- The soil moisture simulation in the third deep layer well captures SMAP behaviors when the SWE exceeds 500 mm. On the other hand, the depth-averaged soil moisture simulation well represent the SMAP observation at the grid cell where SWE does not exceed 300 mm SWE.

- Streamflow predictibilty in the Yellowstone River Basin achieves 0.79 NSE and 0.91 NSE during April to July for the upstream and outlet points compared to 0.50 and 0.73 NSEs. These improvements are associated with the update of the winter precipitation forcing using in-situ snowpack observations.

- The difference of the streamflow amount between the Snake River Basin and the Yellowstone River Basin is attributed to the soil infiltration characteristic such as 'aridic' and 'ustic' for SRB and YRB, respectively. The 'ustic' soil more infiltrates the melt water than the 'aridic' soil in the basin. Thus, the amount of the streamflow in the YRB is more than that of SRB.

This study re-examines approach to updating winter precipitation through the incorporation of in-situ snowpack observations. The focus is on applying a land surface hydrology model to the eastern side of the Continental Divide in the upper Colorado Rockies in North America. The study demonstrates improvements in cold land hydrological processes, encompassing snow, soil moisture, and streamflow. All available hydrologic observations play a crucial role in this process by 1) correcting the weather forcing dataset for snow, 2) assessing surface hydrological processes related to soil moisture, and 3) comparing time-series hydrologic state variables, such as streamflow.



Furthermore, the research emphasizes the significance of underlying soil characteristics in the application of land surface hydrologic models. Even with consistent practices in employing the land surface hydrologic model and utilizing the same

weather forcing dataset, variations in underlying soil characteristics can result in differences in the amount of streamflow at the conclusion of the modelling process. This paper serves as a prototype study of cold land hydrological processes on the eastward side of the Continental Divide in the upper Colorado Mountains. The demonstration utilizes available hydrologic state variables, including snow, soil moisture, and streamflow, within the Yellowstone River Basin.

**Code/Data availability**

The code used to analyze the data and generate the figures in this study is available on the GitHub repository at [https://github.com/UW-Hydro/VIC]. The dataset supporting the conclusions of this article is available in the [https://nsidc.org/data/smap] and [https://nsidc.org/data/au_dysno/versions/1] for soil moisture and snowpack. Researchers interested in accessing the code or data can contact the corresponding author for further information.

**Competing interests**

The author declares competing interests: Dr. Kang is an employee of FedWriters Inc, the company that supports US NOAA. The author affirms that these competing interests did not influence the design, conduct, or reporting of the research findings presented in this paper.

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
