# Peer review of "Characterization of Cold Land Hydrological Processes by Integrating In-Situ Snowpack Observations with a Land Surface Model in the Yellowstone River Basin, USA"

_EGUsphere, 2023_

## Author Comment (AC1)

Dear editor,

In the manuscript "Characterization of Cold Land Hydrological Processes by Integrating In-Situ Snowpack Observations with a Land Surface Model in the Yellowstone River Basin, USA", the author improves simulated streamflow in the Yellowstone river by adjusting the snowfall inputs based on the difference in observed and simulated snow water equivalents. Additionally, the author attributes the streamflow disparity between the Yellowstone and Snake rivers to the difference in soil infiltration in the respective areas. Although the methods used in the manuscript are sound, I would recommend the manuscript to be **rejected** as it is poorly written, contains unsubstantiated conclusions and carries little significance for the scientific community. Below is a more expansive description of my main arguments as well as a list of line-by-line comments.

**➔ Thank you for your comments. The manuscript is now re-written followed by updates to address your line-by-line comments. Responses are in bold and the updated manuscripts are in bold and italics.**

*Scientific significance*

The main body of the manuscript consists of the efforts to improve simulated streamflow in the upper and lower Yellowstone river by adjusting ERA-5 snowfall inputs. As mentioned in the manuscript (lines 226-227), ERA-5 can suffer from snow undercatch (i.e. insufficient snowfall) due to the data's reliance on weather station observations that are often biased towards valleys in mountainous areas. Therefore, the author adjusts the ERA-5 snowfall with a constant factor based on the average difference between the simulated snow water equivalent under a unadjusted ERA-5 simulation and several observations.

However, this approach only confirms what is already known, namely that ERA-5 likely estimates insufficient snowfall in the region. Moreover, the limitations and future use of the approach are not explored. For example, although streamflow simulations are a bit better under the adjusted snowfall amounts (figure 6), the simulated snow water equivalent actually gets worse for several stations (figure 2). As the limitations and usefulness of the approach are not discussed, the manuscript only reflects that streamflow simulations (with a specific hydrological model setup) with more snowfall perform better in the Yellowstone river basin than streamflow simulations without.

**➔ The discussion part is included to address 1) an extension of the correction of the winter precipitation forcing for the existing weather reanalysis dataset and 2) a spatial heterogeneity of the snow water equivalent simulations within the Yellowstone River Basin and beyond the basin.**

*Unsubstantiated conclusions*

There are several issues with the results presented in section 3, that subsequently inform the conclusions in section 4, that I would like to mention. Specifically, they relate to (1) the comparisons between differently calibrated simulations, (2) the comparison between observed and simulated soil moisture and (3) the conclusions drawn from the Yellow river and Snake river simulation differences.

**➔ More details in the Result section were made to specifically distinguish calibration and validation periods of the streamflow performance evaluation for the Yellowstone River Basin. Comparison parts between YRB (Yellowstone) and SRB (Snake) are now moved to discussion. Therefore, the soil moisture evaluation is only for YRB.**

Although the manuscript attributes the streamflow improvements solely to the increase in snowfall, the snowfall adjustment is not the only difference between the simulations. Rather, the manuscript mentions (lines 315-317) that a re-calibration took place between these simulations as well. The impact of this recalibration on the hydrological states and fluxes is never discussed. Therefore it may well be that the streamflow improvement cannot only be attributed to the snowfall adjustment.

**➔ I partly agree that the streamflow calibration anticipates the adjustment of the soil properties. However, the calibration is independently applied to 1) base (without snowfall update) and 2) Mf (with snowfall update). Then, the streamflow performance is evaluated against the observed streamflow. While the soil properties are adjusted to fit the observed streamflow, this equivalent application to both cases shows which one is superior in predicting the streamflow. An additional note is that the Mf soil moisture is more realistic than the base simulation.**

The manuscript also compares the simulated soil moisture with the SMAP satellite observations. However, there a two substantial limitations to this comparison. Firstly, the manuscript only compares simulations and observations on two individual points. I see no reason why a spatial comparison would not be possible and more informative (single points do not accurately reflect model performance). Secondly, the manuscript compares the SMAP signal to the average soil moisture content in either the whole soil column or the third soil layer. However, SMAP only measures the soil moisture content in the upper 5 centimeters of the soil (which is only a part of the first soil layer). Therefore this comparison cannot be valid.

**➔ Spatial comparison of soil moisture is included in Figure 3 (ab) along with explanations. I don't agree SMAP is only valid up to 5 cm. SMAP uses L-band**

**whose penetration depth of the soil column is 50 cm when the soil is dry. Please find the literature about the penetration depth of soil and snow at L-band.**

[Figure]

**Nolan, Matt and Dennis Robert Fatland. "Penetration depth as a DInSAR observable and proxy for soil moisture." *IEEE Trans. Geosci. Remote. Sens.* 41 (2003): 532-537.**

[Figure]

**Casey, J. & Howell, Stephen & Tivy, Adrienne & Haas, Christian. (2016). Separability of sea ice types from wide swath C- and L-band synthetic aperture radar imagery acquired during the melt season. Remote Sensing of Environment. 174. 314-328. 10.1016/j.rse.2015.12.021.**

*It notes that Figure 1ab represents the spatial average of soil moisture between SMAP observation and VIC simulation, where the overall trend confirms a correlation (0.5) between observed and simulated soil moisture, and that high soil moisture is mainly driven by the springtime snowmelt in May and June in the previous year.*

The manuscript also compares the streamflow differences between the Yellow and Snake rivers east and west of the North American Continental Divide. The manuscript concludes that this difference is driven by the higher infiltration rates in the Snake river basin. However, higher infiltration does not directly result in lower streamflow. In fact, the routing scheme used traditionally includes the baseflow, which is essentially infiltrated soil moisture. Therefore, infiltrated water is not lost and cannot be the reason for the streamflow difference.

➔ **You are correct. Baseflow and runoff are merged before transferring into the routing scheme at VIC. Instead of infiltration characteristics, lava soil in the Snake River Basin plays an important role in reducing the streamflow. In the VIC rout scheme, 'Fraction File' is defined. (The fraction file is gridded information about the fraction of each grid cell that flows into the basin being routed.) In the case of the Snake River, 0.2 is inserted to match the amount of the streamflow, but 0.5 is used for the Yellowstone River.  As shown in the figure, 'basalt' soil is dominant in the Snake River Basin, but not in the eastern part of the Continental Divide in the Yellowstone River Basin. This results in the reduction of the streamflow in the Snake River Basin than the Yellowstone River Basin even with a similar level of precipitation.**

[Figure]

**Map created by Zac Lifton.  Geologic units from the *Geologic Map of Idaho* by Lewis et al. (2012; https://www.idahogeology.org/product/m-9)**

*In the VIC rout setting, 0.2 fractional area of the rout scheme is used for the Snake River Basin while 0.5 fractional area is used for the Yellowstone River Basin. As a result, it is recommended to account for different infiltration characteristics when applying land surface hydrology models between the Continental Divide of the upper Colorado Mountains.*

*Writing*

Both the article structure as well as its language could be substantially improved. The structure and language improvements are provided in the line-by-line comments below, but some examples are: there is no discussion, the key knowledge gap (and its references) are located in the results, the approach to adjust the snowfall is introduced

during a data description (and before the model or meteorology description), figures contain errors, code availability reference is incorrect and the comparison between the Yellowstone and Snake rivers has no relation to the rest of the manuscript.

➔ **Manuscript is updated per your comments. It also notes that** *‘2.3 Land Surface Model and Snowfall Update’* **is revised to address the key methodology of the snowfall forcing update using SNOTEL observations.**

*Line-by-line comments*

Line 8 "enhancing streamflow predictability (…) is enabled": How can predictability be enabled? ➔ **The sentence is changed to:**
*In the eastern region of the North American Continental Divide in the upper Colorado Rockies, this study demonstrates that it achieves the enhancement of the streamflow predictability from May to July in the Yellowstone River Basin.*

Line 9-10 "updated winter precipitation weather forcing dataset": "corrected snowfall"➔ **changed to:**
*with the corrected snowfall weather forcing.*

Line 13-14 "streamflow predictability": "streamflow performance"➔ **Changed.**

Line 14: Metrics are given during the calibration period, not the validation period! ➔ **included.**
➔ *0.91, in contrast to the baseline simulation's 0.73 NSE during peak streamflow periods during the calibration period.*

Line 19-22: why these additional results? They seem unconnected to the rest of the manuscript (different region, different research question). ➔ **removed in the abstract.**

Line 19-20 " soil infiltration properties (…) are wetter": How can infiltration properties be wetter? ➔ **removed in the abstract.**

Line 14 "snowpack embraces significant importance": Embraces? ➔ **Changed to:**
*Snowpack plays a significant role in the upper Missouri River Basin situated in the western United States*

Lines 25-40: Strong focus on climate change (and wildfires?) even though the article does not focus on climate change.
➔ **I understand your concerns. But, a limited knowledge of the cold land hydrological processes prevents a better understanding of the recent climate**

**change and unprecedented wildfires in the western United States. In that context, some of the references regarding climate change are included.**

Lines 48-49: "with various researchers establishing this body of knowledge": Would remove ➔ **Removed.**

Lines 55-56: "However, the application of land surface hydrology models to the Yellowstone river (...) is relatively recent": This is not the case, there are many studies that use land surface or hydrological models in this region. An example from the VIC model used in the manuscript is the 1997 paper by Bart Nijssen (Streamflow simulation for continental-scale river basins; DOI 10.1029/96WR03517), where he actually simulates the whole world.
➔ **Included with other references.**
*Hydrologic processes of the Yellowstone were evaluated by Nijseen et al. (1997). However, the application of land surface hydrology models to the Yellowstone River, one of the headwaters of the Upper Missouri River, is still limited (Kannan et al. 2019; Flemming et al. 2021).*

Line 58 "the understanding of hydrological processes remains limited": Use the word "understanding" with care, as model simulations also do not directly contribute to "understanding". Rather, they use the current understanding to predict trends/changes. ➔ **Updated:**
*the current understanding of evaluation for hydrological processes in the headwater watershed remains limited.*

Section 1.1: Generally too much detail about the region and the streamflow gauges, but no map. ➔ **Introduction of Figure 1 is included.**
*Detailed descriptions of these streamflow gauges are summarized in Table 1, and Figure 1 depicts the Yellowstone River Basin with locations of the streamflow gauges.*

Line 106 "fo": "for"➔ **I cannot find it.**

Line 139-141: Why do you assume the meteorology is wrong and not the model**?==> The sentence is removed.**

Lines 155-156: "The VIC model is configured with 18 rows and 29 columns": This provides no information to the reader ➔ **removed.**

Lines 162-163 "determined using the minimum and maximum elevations in the watershed": Determined how? ➔ **Changed to:**
*0.5 to 3.5 based on $VIC_{BL}$, varied with elevations in the watershed.*

Line 168 "ROUT": Newly introduced but never mentioned or explained. ➔ **changed to:**
*VIC and a separate routing scheme (ROUT)*

Lines 191-193: Repeat of information. ➜ **Removed the sentence, "Streamflow accuracy is assessed at two USGS streamflow gauge points in Livingstone and Billings, MT, USA. However, due to spatial and temporal variations in soil moisture across the Yellowstone River Basin, spatio-temporal observations are necessary."**

Line 199 "In cases of deep (above 200mm SWE) and shallow (below 200mm SWE)": So in all cases. ➜ **The sentence is now changed to:**
*With two categories of SWE with 200 mm criteria, three-layered soil moisture simulations are compared against a single SMAP value.*

Section 3: Generally too much new information, which should have been given in the introduction or methods, in the results. Moreover, there is a strong need for quantification of the results (KGE, correlation, mean squared error and more).
➜ **Table 3 is now updated with NSE, Mean Absolute Error, and Root Mean Squared Error.**

Line 209 "Following the methodology of the prior study conducted by Kang and Jung (2023)": What methodology?==> **changed to:**
*Following the snowfall update methodology of the prior study conducted by Kang and Jung (2023)*

Line 225 "European Reanalysis Assimilation 5 (ERA-5)": "ECMWF Reanalysis v5 (ERA5)"!
➜ **corrected.**

Line 258-259 "both SWE and soil moisture simulations align well with in-situ SWE and satellite-based soil moisture observations": I do not agree, performance is rather poor.
➜ **Updated.**
*general trends of both SWE and soil moisture simulations align well with in-situ SWE and satellite-based soil moisture observations*

Line 321: " VICMF (0.86) (...) is outperformed by VICBL (0.58)": Does a higher NSE not indicate that VICMF it performs better?=> **Updated.**
*In Table 3, for the upstream, $VIC_{MF}$ (0.86) during calibration outperformed $VIC_{BL}$ (0.58), and during the validation period, it exhibited a better $VIC_{MF}$ (0.73) than $VIC_{BL}$ (0.54).*

Line 326-327 "is more pronounced at the upstream than in the outlet": results show exactly the opposite, with a larger NSE improvement in the upstream than at the outlet.==> **Updated.**
*The impact of the snowfall updates is more pronounced at the upstream point than at the outlet, showcasing a more consolidated effect.*

Table 3 and 4: Why are performance metrics selected during the calibration period? This should be during the validation period. ➜ **Corrected.**

Lines 331-332: "specifically, in 2002, 2008 and 2018": Why select these years**?==> This selection of water years is intended to demonstrate 'small', 'medium', and 'large' amounts of streamflow.**

Figure 6: three timeseries where there should be two. Middle is duplicate of top. ➔ **corrected.**

Lines 386-391: Why not directly compare the soil parameters in the input? ➔

[Figure]

**Snake River Basin**                    **Yellowstone River Basin**

**As the figures above show, the left is from the Snake River Basin on the western side of the continental divide and the right is from the Yellowstone River Basin on the eastern side. As you can see, the magnitude is about 70 times in the Yellowstone River basin to the Snake River Basin. The current VIC version doesn't consider the groundwater component, and the Snake River Basin is geologically volcanic soil that infiltrates more water to the groundwater. In the VIC setting, the fractional area of the grid cell as of 0.2 is adjusted in Snake River application to fit the streamflow. However, the Yellowstone River Basin has 0.5 for the fractional area of the grid cell in the Rout scheme. Therefore, the comparison of the soil inputs is not necessary in this context.**

Lines 408-409: This is not a conclusion. ➔ **removed.**

Lines 436-437: This GitHub repository only contains the model code, not the "code used to analyze the data and generate the figures in this study"! ➔ **only VIC source codes are publicly available. The sentence is changed to my Matlab codes.**
***Anyone interested in accessing and visualizing the code or data can contact the corresponding author for further information.***

Sections headers are incorrect (twice section 3 and 3.2) ➜ **Updated.**

No discussion! ➜ **Discussion is included with 4 elements.**

***When evaluating hydrological processes in the Yellowstone River Basin, the paper finds four elements to be discussed: 1) a snow undercatch when measuring snowfall from a traditional tipping bucket, 2) a comparison of the hydrologic responses with a continental divide in the Upper Colorado Mountains, 3) a comparison of the hydrologic responses between upstream and downstream within the Yellowstone River Basin, and 4) a snowmelt-runoff influence on the soil moisture in the snowmelt-dominant watershed.***